# MULTI-REPRESENTATION ENSEMBLE IN FEW-SHOT LEARNING

## ABSTRACT

Deep neural networks (DNNs) compute representations in a layer by layer fashion, producing a final representation at the top layer of the pipeline, and classification or regression is made using the final representation. A number of DNNs (e.g., ResNet, DenseNet) have shown that representations from the earlier layers can be beneficial. They improved performance by aggregating representations from different layers. In this work, we asked the question, besides forming an aggregation, whether these representations can be utilized directly with the classification layer(s) to obtain better performance. We started our quest to the answer by investigating the classifiers based on the representations from different layers and observed that these classifiers were diverse and many of their decisions were complementary to each other, hence having the potential to generate a better overall decision when combined. Following this observation, we propose an ensemble method that creates an ensemble of classifiers, each taking a representation from a different depth of a base DNN as the input. We tested this ensemble method in the setting of few-shot learning. Experiments were conducted on the mini-ImageNet and tiered-ImageNet datasets which are commonly used in the evaluation of few-shot learning methods. Our ensemble achieves the new state-of-the-art results for both datasets, comparing to previous regular and ensemble approaches.

## 1 INTRODUCTION

The depth of a deep neural network is a main factor that contributes to the high capacity of the network. In deep neural networks, information is often processed in a layer by layer fashion through many layers, before it is fed to the final classification (regression) layer(s). From a representation learning point of view, a representation is computed sequentially through the layers and a final representation is used to perform the targeted task. There have been deep neural networks that try to exploit the lower layers in the sequence to achieve better learning results. GoogLeNets (Szegedy et al., 2015) added auxiliary losses to the lower layers to facilitate training. Skip links (such as the ones used in ResNet (He et al., 2016) and DenseNet (Huang et al., 2017)) may be added to connect the lower layers to the higher ones in a deep architecture. Even though the main purposes of these approaches are to assist the training process or to help the gradient back-propagation, the success of these approaches suggests that the representations from the lower layers may be beneficial to many learning tasks. Therefore, it is worth to rethink the standard sequential structure where a final representation is used to make the prediction. In this work, we ask the question whether the representations from the lower layers can be used directly (instead of being auxiliary or being aggregated into a final representation) for decision making. If so, how can we take advantage of these lower-level representations and what are good practices in doing so?

We first investigated the problem by conducting classifications using the representations from different layers. We took the convolutional layers of a trained network as an encoder. The representations (feature maps) from different layers of the encoder were tested for their classification performance. We observed that although overall, the feature maps from the higher layers led to better performance, there was a significant number of cases that correct predictions could be made with the lower feature maps but the higher-level feature maps failed to do so. This suggested that the lower-level representations have the potential to help the classification directly (detailed analysis in Section 3).

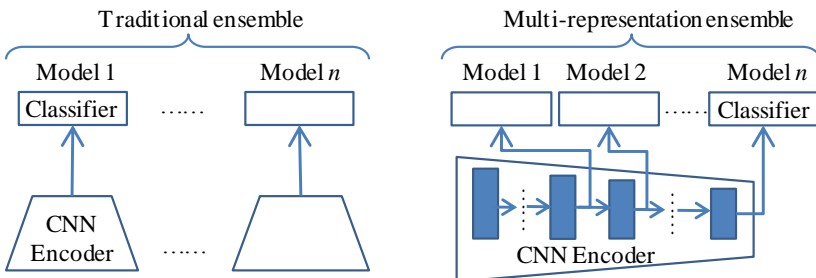

Figure 1: Multi-representation classifier ensemble.

Based on the inspiration from the prior models (i.e., GoogLeNet, ResNet and DenseNet) and our own observations, we propose an ensemble approach to directly take advantage of the lower-level representations. By integrating multiple models, an ensemble is likely to compensate the errors of a single classifier, and thus the overall performance of the ensemble would be better than that of a single classifier. This makes ensemble a suitable technique for our purpose. A variety of methods exist for ensemble construction. Some utilize sampling to obtain individual models from different subsets of the training data. Others construct models with different structures or initialization. However, these common ensemble methods cannot achieve our goal to exploit the lower-level representations. Instead, we propose a special type of ensembles, different from the existing ones. In particular, each classifier in our ensemble takes a feature map from a different depth of a CNN encoder as input and the whole ensemble utilizes the feature maps from multiple convolutional layers. We call this approach the *multi-representation ensemble*. Figure 1 illustrates our ensemble approach and compares it to the common ensemble method.

We evaluate our ensemble method on the few-shot learning (FSL) problem (Snell et al., 2017). FSL aims to learn a network capable of recognizing instances (query images) from novel classes with only few labeled examples (support images) available in each class. Given the demanding nature (learning from a few examples) of the problem, many FSL approaches first train an encoder following a regular training paradigm and then further-train the encoder and the classifier using a FSL paradigm. Because the encoder plays an important role in few-shot learning, it is a good learning task to apply and test our ensemble method which takes advantage of multiple representations from the encoder. Note that in recent years, many FSL works have employed extra data from the test (novel) classes for better performance. The extra data can be unlabeled and given at the test time (transductive learning) (Kye et al., 2020; Yang et al., 2020) or during the training phase (semi-supervised learning) (Rodríguez et al., 2020; Lichtenstein et al., 2020). Our problem scope focuses on the traditional FSL setting, where only a few (one or five) support images per novel class are available at the test time.

Experiments with our ensemble model were conducted on two FSL benchmark datasets and we obtained new state-of-the-art results for both. Besides evaluating our ensemble and comparing it to the existing methods for FSL tasks, we also conducted experiments that demonstrated that the utilization of multiple representations in the ensemble is crucial for the success of our method. Our main contributions are as follows: 1) We propose a novel ensemble method that creates a collection of models by employing multiple representations from different depth of a deep neural network. 2) We demonstrated the advantage of our ensemble model on the FSL problems and achieved new state-of-the-art results on two benchmark datasets. Our experiments also showed that multi-representation is necessary for the improved performance of the ensemble.

## 2 RELATED WORK

**Ensemble methods**. Ensemble methods are commonly used to improve prediction quality. Some example ensemble strategies include: (1) manipulate the data, such as data augmentation or dividing the original dataset into smaller subsets and then training a different model on each subset. (2) apply different models or learning algorithms. For example, train a neural network with varied hyperparameter values such as different learning rates or different structures. (3) hybridize multiple ensemble strategies, e.g., random forest. Ensembles have also been applied to FSL problems. Liu et al. (2019b) proposed to learn an ensemble of temporal base-learners, which are generated along

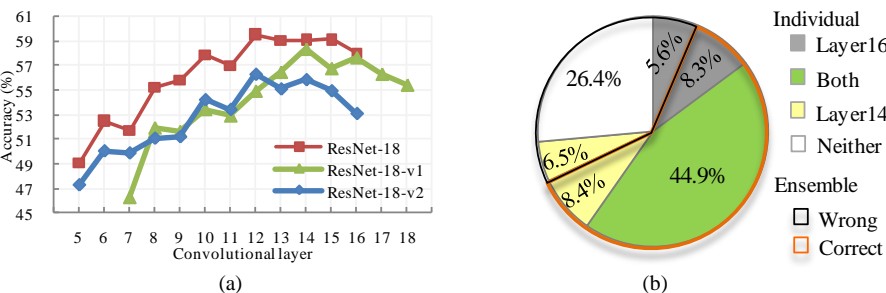

Figure 2: (a) 5-way, 1-shot classification accuracy on mini-ImageNet across convolutional layers. (b) Classification results for convolutional layer 14 and 16 of ResNet-18 and the ensemble incorporating both.

the training time, producing encouraging results on the mini-ImageNet and the Fewshot-CIFAR100 datasets. Dvornik et al. (2019) introduced mechanisms to encourage cooperation and diversity of individual classifiers in an ensemble model. The main difference between our method and the previous ones is the ensemble construction that utilizes multiple representations from different depth of a neural network.

**Few-shot learning**. Meta-learning method has shown great success in FSL (Finn et al., 2017; Grant et al., 2018; Lee & Choi, 2018). MAML (Finn et al., 2017) used a meta-learner that learns from the training images to effectively initialize a base-learner for a new learning task in the test dataset. Further works aimed to enhance the generalization ability by improving the learning algorithm (Nichol et al., 2018), fine-tuning the image embedding space (Sun et al., 2019; Rusu et al., 2018). Another popular direction for FSL is metric-learning which targets on learning metric space where classes can be easily separated. For example, Prototypical Networks use euclidean distance with each class prototype set to be the mean of the support embeddings (Snell et al., 2017). Relation network (Sung et al., 2018) was proposed to compute the similarity score between a pair of support and query images. The query image is classified into the category with the highest similarity. Each individual model in our ensemble employs a relation network for classification while the relational network in different models receives different representation as input. Many recent FSL studies proposed approaches that utilized extra unlabled data or other additional information (Li et al., 2019b; Kye et al., 2020; Yang et al., 2020; Hu et al., 2020; Rodríguez et al., 2020; Lichtenstein et al., 2020). They are not in the scope of the problem we were considering and thus not compared in the result section.

## 3 MOTIVATION

As a beginning investigation on the lower layer representations, we conducted a set of experiments to gain insight into their classification power. We took the FSL setting as the experiment environment and measured the performance of the representations from different convolutional layers in the encoder. (ResNet-18 and its two variants were used as encoders. After encoder pretraining, FSL was conducted using the representation from an encoder layer combined with a classification network. Representations from different layers were tested.) Figure 2(a) shows the classification accuracy across different convolutional layers. The best performance did not come from the final convolutional layers (layer 18 in ResNet-18-v1 and 16 in ResNet-18 and ResNet-18-v2). Instead the fourth or the fifth layer from the last (layer 14 in ResNet-18-v1 and 12 in ResNet-18 and ResNet-18-v2) generated the highest accuracy. Going further towards the lower layers, we observe that overall for the three models, the lower the convolutional layer is, the worse the classification performance becomes.

We looked further into the predictions made by different layers and investigated whether these predictions can be complementary, that is, is there enough diversity among the predictions such that by combining them together, we may obtain a better prediction. Taking layers 16 and 14 of ResNet-18 as examples, we examined the FSL predictions on a random sample of 75,000 images. The classification results using the representations from these two layers are shown in Figure 2(b). The model

using the representation from layer 14 had a little higher overall accuracy ((44.9 + 8.4 + 6.5)% = 59.8%) than that of the one using layer 16 ((44.9 + 5.6 + 8.3)% =58.8%). We can categorize the results into 4 scenarios: 1) Both classified correctly (labeled and colored as "Both" in Figure 2(b)); 2) Only the model using representation from layer 16 made correct prediction (labeled and colored as "Layer16"). The model with layer 14 gave incorrect results to these images; 3) Only the model with layer 14 predicted correctly (labeled and colored as "Layer14"); 4) Neither made correct prediction (labeled and colored as "Neither"). Although many images were classified correctly by both models (44.9% of the total), some images could be only recognized by one of them (scenario 2 and 3).

In an ideal situation, if we find a way to resolve perfectly the conflicts between the two models in both scenarios 2 and 3 and make correct prediction for these images, a classification accuracy as high as 73.7% may be achieved. Ensemble is a potential approach towards this goal. We did a quick test by constructing an ensemble from the two models and using the average of their outputs as the output of the ensemble. Figure 2(b) also shows the result of this quick ensemble. The ensemble reached a accuracy of 61.6%, higher than any individual model. Clearly, for images in scenario 1, the ensemble still made the correct predictions. For images in the scenarios 2 and 3, the ensemble was able to resolve more than half of them and make correct classifications. (There were 13.9% of the images in scenario 2, 8.3% were resolved and had correct predictions from the ensemble. There were 14.9% of the images in scenario 3, 8.4% were resolved.) Although the ensemble did not reach the ideal limit mentioned above, it did provide a better classification, more accurate than any of the individual model in the ensemble. This quick experiment shed light on the possibility to exploit representations from different layers in a deep neural network as a way to obtain better learning performance. Following this direction, we designed our ensemble method for FSL tasks. We conducted more thorough experiments to validate the approach and to determine the design choices that can optimize the performance gain.

# 4 METHODOLOGY

## 4.1 PROBLEM DEFINITION

Formally, we have three disjoint datasets: a training set $D_{\text{train}}$, a validation set $D_{\text{val}}$, and a testing set $D_{\text{test}}$. In the traditional $C$-way $N$-shot classification in FSL, we are tasked to obtain a model that can perform classification among $C$ classes (in the testing set $D_{\text{test}}$) while we have access to only $N$ samples from each class. (The model can be pre-trained using data in $D_{\text{train}}$ or even data in $D_{\text{val}}$. However, there are no overlapping classes between $D_{\text{test}}$ and $D_{\text{train}}$ or $D_{\text{val}}$.) The set of $C \times N$ samples is often referred to as the support set. In many FSL works, $N$ is commonly set to be 1 or 5. We remark that some recent FSL researches have started to explore helps from additional information. In particular, a number of methods have been proposed to leverage unlabeled data beyond the $N$ examples from the $C$ classes to enhance model accuracy. They either use the unlabeled data in the training (semi-supervised learning) or perform classifications with a set of query data together (transductive learning) (Li et al., 2019b; Kye et al., 2020; Yang et al., 2020; Hu et al., 2020; Rodríguez et al., 2020; Lichtenstein et al., 2020). In this work, we focus on the traditional FSL setting where no additional information or unlabeled data are available and the prediction for a query data point is made independently from (without knowing) any other query data points.

## 4.2 MULTI-MODEL MULTI-REPRESENTATION ENSEMBLE

Our ensemble contains multiple encoders (encoders of different network structures). Hence the ensemble is also multi-model. An encoder generates multiple representations. Each is then fed to a classifier network. The combination of a particular representation and a classifier network forms an individual classification model in the ensemble. Figure 3 shows the overall architecture of the ensemble. (For simplicity, the figure illustrates only one encoder with multiple representations. An actual ensemble may include several encoders.) In the following, We describe the details of the ensemble.

Same as many other research work on FSL, we consider the task of image classification. Therefore $D_{\text{train}}$, $D_{\text{val}}$, and $D_{\text{test}}$ are collections of images and their labels. Before being employed in an ensemble to provide multiple representations, an encoder is pre-trained following the pretraining protocol in FSL (Rusu et al., 2018; Sun et al., 2019). If multiple encoders are employed, each of them

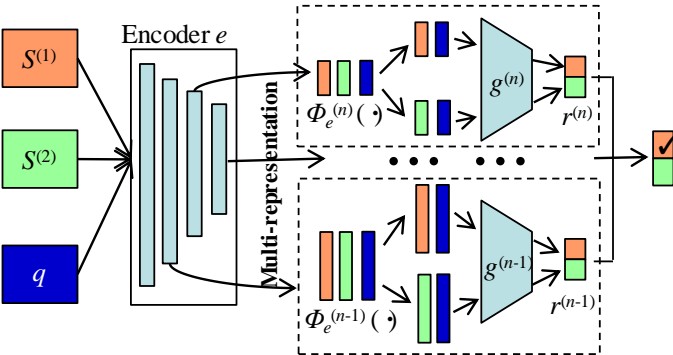

Figure 3: Classification by multi-representation ensemble for a 2-way 1-shot problem.

will undergo the pretraining independently. After the pretraining stage, we have a set of encoders and each encoder can provide multiple representations (image features) from different layers. We refer to the features for image $x$ from $n$-th convolutional layer in the encoder $e$ by $\phi_e^{(n)}(x)$. Our ensemble uses relation network (Sung et al., 2018) to perform classification. The relation network takes a pair of instances from a particular representation as input and outputs a similarity score. We denote by $g$ the score function computed by the relation network.

Let $S^{(1)}$ and $S^{(2)}$ be two images from two different classes (class 1 and 2) in the support set. Let $q$ be a query image to be classified. The images are first fed into the encoders and for each image, multiple representations (from different encoders and different layers) are generated. Take an encoder $e$ and the $n$-th layer of $e$ as an example, We illustrate the classification process using the representations $\phi_e^{(n)}(s^{(1)})$ (orange in Figure 3), $\phi_e^{(n)}(s^{(2)})$ (green in Figure 3) and $\phi_e^{(n)}(q)$ (blue in Figure 3). (Note that Figure 3 contains two sets of representations $\phi_e^{(n)}(\cdot)$ and $\phi_e^{(n-1)}(\cdot)$ with the same color scheme. Colors are used to indicate the source image from which the representation is computed. They are not related to the location, i.e., the encoder and the layer, where the representation is generated.) The representation of $\phi_e^{(n)}(q)$ is concatenated (on the channel dimension) to the representation of $\phi_e^{(n)}(s^{(1)})$. The relation network is applied to the concatenation and a score $r$ is calculated to measure the similarity between the query and the class (class 1) representative image $s^{(1)}$, i.e. $r^{(1)}(q) = g(\phi_e^{(n)}(q)||\phi_e^{(n)}(s^{(1)}))$ where $||$ indicates concatenation on the channel dimension. In an $N$-shot ($N > 1$) setting, for each class, we have $N$ examples (e.g., $S_1^{(k)}, S_2^{(k)}, \ldots, S_N^{(k)}$ for class $k$). A prototype is obtained by averaging the representations $\phi_e^{(n)}(s_i^{(k)})$. The score is computed using the concatenation of $\phi_e^{(n)}(q)$ and the prototype, i.e., in general,

$$r^{(k)}(q) = g(\phi_e^{(n)}(q)||\frac{1}{N}\sum_i \phi_e^{(n)}(s_i^{(k)})) \tag{1}$$

This calculation is performed for each class to produce $C$ scores $r^{(1)}, r^{(2)}, \ldots, r^{(C)}$ for a $C$-way classification. The query image is classified into the category that has the maximum score.

The above presents the classification process of an individual model in the ensemble. There are a set of encoders $e_1, e_2, \ldots$ and for each encoder $e_i$, multiple representations from different layers $n_1^{(e_i)}, n_2^{(e_i)}, \ldots$ are used to produce models in the ensemble. Let $T = \{n_j^{(e_i)}\}$ be the set of representations, each leading to a classifier. Let $r_t^{(i)}(q), t \in T, i \in \{1, 2, \ldots, C\}$ be the score for image $q$ with respect to class $i$ and computed by the classifier using representation $t$. The ensemble's final classification is made following the average scores over the models, i.e., the class label $c^*(q)$ is predicted to be:

$$c^*(q) = \arg\max_i \frac{1}{|T|}\sum_{t \in T} r_t^{(i)}(q) \tag{2}$$

where $|T|$ is the size of the set $T$. Each relation network in the ensemble will be trained independently following the training procedure in (Sung et al., 2018) while the encoders remain unchanged after pre-training.

# 5 EXPERIMENT RESULTS

## 5.1 EXPERIMENTAL SETUP

**Datasets**. We used two standard benchmark datasets in FSL: mini-ImageNet (Vinyals et al., 2016) and tiered-ImageNet (Ren et al., 2018). Mini-ImageNet consists of 100 categories, 64 for training, 16 for validation and 20 for testing, with 600 images in each set. Tiered-ImageNet includes 608 classes (779, 165 images) split as 351 training, 97 validation and 160 testing classes, each with about 1300 images. The image size of both datasets is $84 \times 84$.

**Implementation details**. We adopt the same ResNet-18 model in (Han et al., 2020) and two variants of ResNet-18 as the encoders. The first variant is ResNet-18-v1 in (Sun et al., 2019) which has 3 residual sections with 3 basic blocks, each including 2 convolutional layers, and the second one is ResNet-18-v2 consisted of 2 residual sections with 4 basic blocks. In order to pretrain an encoder that produces image features/representations, the first step is similar to the common pre-training stage used in FSL (Rusu et al., 2018; Sun et al., 2019). We merged data of all classes in $D_{\text{train}}$ for pre-training. After pre-training, we added shift and scaling parameters for the convolutional layers in the encoder and trained the parameters by the MTL approach used in (Sun et al., 2019). To further improve generalization, we also fine-tuned the upper layers in each encoder using $D_{\text{val}}$ as unlabelled data, following the method proposed in (Han et al., 2020). Specifically, we fine-tuned the upper half layers of the encoder while freezing the rest. After pre-training, the encoder remains unchanged.

The evaluation of the ensembles was conducted using $C$-way $N$-shot classification tasks with $C = 5$ and $N = 1$ or $N = 5$. In an episode of evaluation, a classification task was constructed by randomly selecting $C$ classes and $N$ samples per class from $D_{test}$ to serve as the support set. 15 random query data points from the $C$ classes were also selected as the query images to test the classification. The evaluation process consisted of 1000 episodes. The mean accuracy (in %) over the 1000 episodes and the 95% confidence interval are reported in the experiment results. Input images were re-scaled to the size $80 \times 80$ and normalized before fed into the model. The relation networks to produce the similarity scores between the support and the query images were composed of 2 convolutional layers followed by a fully-connected layer with a sigmoid function. Our implementation was based on PyTorch (Paszke et al., 2017a). Pre-training phase used default parameters in the works we cited. For optimization of the relation networks , we used stochastic gradient descent (SGD) with the Nesterov momentum 0.9. The initial learning rate was set to 1e-2.

## 5.2 MAIN FSL RESULTS

The ensemble used in this section employed representations from the last (from top) 9 convolutional layers in ResNet-18 and the last (from top) 6 layers in ResNet-18-v1. The results on the two benchmark datasets are shown in Table 1. As stated in previous sections, we focused on traditional FSL. In Table 1, we compare our results only to the best prior results on traditional FSL. Table 1 shows that our model gives the new state-of-the-art performance on the 1-shot and the 5-shot tasks for both the mini-ImageNet and the tiered-ImageNet datasets. This illustrates the effectiveness of the multi-model multi-representation ensemble for the FSL problems.

## 5.3 ABOLITION STUDY AND ENSEMBLE DESIGN

In this section, we further demonstrate the benefit of lower-level features using results from the abolition experiments and discuss the design choices for constructing a better ensemble. For abolition study, we compared ensembles that utilized different sets (subsets) of representations/encoders. We first present and discuss the performance results from ensembles that involve multi-representation from a single encoder and then continue to ensembles that employ both multiple representations and encoders. The ensemble performances (accuracy) were measured for few shot classification on the mini-ImageNet dataset.

| Model | Encoder | mini-ImageNet | | tiered-ImageNet | |
|---|---|---|---|---|---|
| | | 1-shot | 5-shot | 1-shot | 5-shot |
| TADAM (Oreshkin et al., 2018) | ResNet-12 | $58.50 \pm 0.30$ | $76.70 \pm 0.30$ | - | - |
| MTL (Sun et al., 2019) | ResNet-12 | $62.10 \pm 1.80$ | $78.50 \pm 0.90$ | $67.8 \pm 1.8$ | $83.0 \pm 0.7$ |
| TapNet (Yoon et al., 2019) | ResNet-12 | $61.65 \pm 0.15$ | $76.36 \pm 0.10$ | $63.08 \pm 0.15$ | $80.26 \pm 0.12$ |
| MetaOpt-SVM (Lee et al., 2019) | ResNet-12 | $62.64 \pm 0.61$ | $78.63 \pm 0.46$ | $65.99 \pm 0.72$ | $81.56 \pm 0.53$ |
| CAN (Hou et al., 2019) | ResNet-12 | $63.85 \pm 0.48$ | $79.44 \pm 0.34$ | $69.89 \pm 0.51$ | $84.23 \pm 0.37$ |
| CTM (Li et al., 2019a) | ResNet-18 | $64.12 \pm 0.82$ | $80.51 \pm 0.13$ | $68.41 \pm 0.39$ | $84.28 \pm 1.73$ |
| **Ensemble** | | | | | |
| Robust-dist++ (Dvornik et al., 2019) | ResNet-18 | $59.48 \pm 0.62$ | $75.62 \pm 0.48$ | - | - |
| MTL+E$^3$TB* (Liu et al., 2019a) | ResNet-25 | 64.3 | 81.0 | 70.0 | 85.0 |
| Ours | ResNet-18 | **65.01±0.66** | **82.31±0.49** | **70.87±0.57** | **85.31±0.31** |

Table 1: The 5-way, 1-shot and 5-shot classification accuracy (%) on mini-ImageNet and tiered-ImageNet datasets. Average classification performance over 1000 randomly generated episodes, with 95% confidence intervals. (* Confidence intervals are not reported in the original paper.)

**Single Encoder Multiple Representation**. Figure 2(a) shows that the best accuracy obtained by a classifier using a single representation from ResNet-18 is 59.53% (with representation from layer 12). In Table 2, we observe that ensembles utilizing multi-representation have boosted performance. For example, the ensemble containing classifiers using representations from layer 16 to layer 14 of ResNet-18 achieves an accuracy of 62.59% (row 1 of Table 2). Including more classifiers that use representations from layers 13, 12 and 11, the ensemble can reach an even higher performance (63.54%, row 2). The topmost accuracy (64.03%) from the ensembles based on ResNet-18 comes from the one that utilizes representations from layers 16 to 8. The improvement by including multiple representations into the ensemble can also be observed for the ResNet-18-v1 and the ResNet-18-v2 encoders. In general, for single encoder ensembles, coming down from the top layer, the more representations we include in the ensemble, the better performance the ensemble can achieve, until a turning-point layer is reached. Adding representations after the turn point may lead to reduced performance.

**Multiple Encoder Multiple Representation**. Incorporating different models (e.g., neural networks of different structures) is a common method to construct an ensemble. Clearly, this multi-model construction can be combined with our multi-representation construction to create ensembles that employ both multiple models and multiple representations from each model. The last row of Table 2 shows the configuration and the performance of one of our multi-model, multi-representation ensembles used in the experiments. There is a significant performance gain comparing this ensemble to those on row 13 and 14. The ensembles on row 13 and 14 are multi-model but not multi-representation since only one representation is used from each of the encoders in the ensembles. Although a multi-model ensemble already performs better than the individual models in the ensemble, adding multi-representation on top of multi-model leads to even better performance, giving rise to the best performer among the ensembles (row 15).

**Selection of Encoders and Representations for the Ensemble** Not all encoders or representations are helpful for an ensembles. Some representations (such as the ones from layers 7 and 6 of ResNet-18, layers 12 and 11 of ResNet-18-v1), when incorporated into the ensemble, led to performance decline (rows 4 and 8 in Table 2 comparing to rows 3, 7). We observe the classifiers that employed these individual representations also showed lower performance (Figure 2(a)). Similarly, the performance of a multi-encoder ensemble may degrade considerably if an encoder that falls behind the others is included. Our final ensemble did not include ResNet-18-v2 as an encoder because its performance is significantly lower than that of ResNet-18 and ResNet-18-v1. To select an encoder or a representation to be included in an ensemble, we test, under the same task using the validation data, the performance of the individual encoders and representations. (For an encoder, we use the best performance across its layers as the measure for the encoder.) A relative threshold $0 < \tau < 1$ is established. An encoder (representation) is selected if its performance is above $\tau$ of the performance of the best encoder (the best representation from the same encoder). We set $\tau = 0.93$ to construct our final ensemble.

**Ensemble vs Aggregated Representation**

| Row# | Encoder(s) | Conv layers | Accuracy(%) |
|------|-----------|-------------|-------------|
| 1 | ResNet-18 | 16–14 | 62.59 |
| 2 | ResNet-18 | 16–11 | 63.54 |
| 3 | ResNet-18 | 16–8 | 64.03 |
| 4 | ResNet-18 | 16–6 | 63.71 |
| 5 | ResNet-18-v1 | 18–17 | 58.57 |
| 6 | ResNet-18-v1 | 18–15 | 59.67 |
| 7 | ResNet-18-v1 | 18–13 | 60.72 |
| 8 | ResNet-18-v1 | 18–11 | 59.21 |
| 9 | ResNet-18-v2 | 16–14 | 56.05 |
| 10 | ResNet-18-v2 | 16–12 | 57.49 |
| 11 | ResNet-18-v2 | 16–10 | 58.26 |
| 12 | ResNet-18-v2 | 16–8 | 58.77 |
| 13 | ResNet-18, ResNet-18-v1 | 16, 18 | 60.87 |
| 14 | ResNet-18, ResNet-18-v1, ResNet-18-v2 | 16, 18, 16 | 58.99 |
| 15 | ResNet-18, ResNet-18-v1 | 16–8, 18–13 | 65.01 |

Table 2: Classification accuracy of multi-model multi-representation ensembles. (The second column lists encoders and the third lists layers (or ranges of layer) from which representations were used. If there were multiple encoders, their layer(s) information is separated by a comma.)

As discussed in the introduction, one of the motivations for us to explore representations from different layers is the skip links in neural networks such as DenseNet. We conducted experiments to compare our multi-representation ensemble to an alternative that uses DenseNet-like aggregated representation. In the alternative approach, representations from different layers were aggregated, by concatenation, into a single representation and classifications, using the relation network, were performed on this single representation. The results of comparison are shown in Figure 4. We tested these two methods on the mini-ImageNet dataset in a 5-way, 1-shot setting. The performance of our multi-representation ensemble is much better than that of the aggregated representation alternative. Moreover, aggregating more representations does not necessarily benefit the classification since the model accuracy slightly drops and then stands still as representations from more layers are added in. This shows that not all approaches which try to utilize multiple representations can attain performance gain from these representations. Our multi-representation ensemble is one approach that is highly effective in taking advantage of the multiple representations.

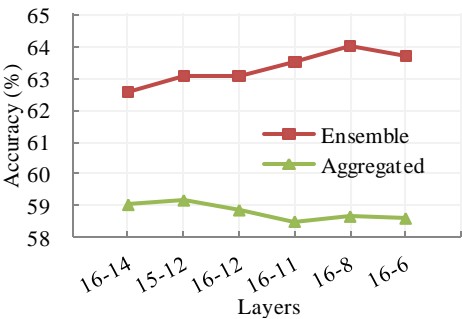

Figure 4: Comparison of classification accuracy between multi-representation ensemble and aggregated representation. Marks on the x-axis indicate the range of convolutional layers.

## 6 CONCLUSIONS

In this paper we propose a new ensemble method that creates an ensemble of classifiers, each using the representation/feature map from a different depth in a CNN encoder. Through experiments, we validated the effectiveness of our model on FSL problems. Our ensemble achieved the new state-of-the-art results on two commonly-used FSL benchmark datasets. We further conducted experiments and analysis to investigate the selection of representations for creating a better ensemble. It is quite possible that the multi-representation approach is not limited to the few-shot learning problems. As future work, we plan to explore more scenarios where multi-representation can be applied. Furthermore, the multi-representation approach does not have conflict with many other ensemble methods. It can be combined with other types of ensemble constructions to produce a better ensemble.

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
