# OpenReview forum: "Multi-Representation Ensemble in Few-Shot Learning"
_ICLR.cc/2021/Conference — Reject_

### Official Review · AnonReviewer4 · 2020-10-24
**Contribution is rather technical and comparison to the state of the art is incomplete**

**Rating:** 4
**Confidence:** 4

**Review:**

The authors propose a simple approach, which obtains competitive results with the state of the art of few shot learning. However, I have the following concerns:

- the proposed method is somewhat incremental. The authors propose to average the predictions of classifiers that take as input different features from the backbone.
- While it’s a sensible thing to try, in my understanding, the proposed method is equivalent to a simpler approach, that would simply concatenate those features and learn a classifier on the concatenated features. I believe that this approach, and a number of other simple baselines employing a wider representation space extracted from the backbone, would be important to strengthen the analysis of the proposed method.
- The presentation of state of the art results is incomplete. Dvornik et al. also report results for tiered-Imagenet (which surpass the reported results). Some other relevant works would need to be cited and compared to [1, 2] (some of their results also surpass the reported results.)
- Organisation: method and results should be presented separately: the current flow of the paper alternates between empirical findings (motivation), a formal approach (methodology) and experimental results. This structure suggests that the submission would likely be better suited for a more technical venue. It would also better to isolate in a background section the presentation of the baseline approach (Sung et al. 2018), before presenting the proposed method itself, to make it more evident what the contributions are.
- the authors do not motivate the chosen experimental setting (FSL): there is no analysis of why the proposed approach should be particularly well suited to address the specificities and challenges of this task.
- it seems to me that employing so many linear classifiers (on increasingly larger dimensional features) would lead to a large increase in parameter count - but the authors do not perform any analysis regarding this aspect.
- Overall, a lot of polishing of the paper is needed prior to publication. Please find a few comments in that respect below.

Comments:
- Figure 2: what is ResNet-18 (in red) if it’s not v1 or v2 ? On this note, both papers should be cited when they are introduced in Section 3 (only ResNet v1 is cited.)
- “Our ensemble contains multiple encoders (encoders of different network structures).” At this point, this is not very clear: is the method used on top of a traditional ensemble ?
- MULTI-MODEL MULTI-REPRESENTATION ENSEMBLE sounds tautological.
- abolition should probably be ablation


[1] Few-Shot Learning via Embedding Adaptation with Set-to-Set Function, Ye et al. CVPR 2020
[2] Adaptive Subspaces for Few-Shot Learning Simon et al CVPR 2020

---

### Official Review · AnonReviewer1 · 2020-10-25
**This paper has good motivation review while is lack of scientific analysis and results support**

**Rating:** 5
**Confidence:** 3

**Review:**

Thanks to the authors for providing such an ensemble approach. This paper aims to find a way to directly utilize representations with the classification layer(s) to obtain better performance.  The ensemble method is able to create an ensemble of classifiers. And the ensemble achieves the new state-of-the-art results in a few-shot setting, comparing to previous regular and ensemble approaches.

This topic is very straightforward and would be very easy for the audience to understand. While the results might not be that convincing enough. The biggest concern is the contribution of this paper, to be more specific, the proposed method might not be useful and might need to be tuned in other few-shot settings. The mini-ImageNet and tiered-ImageNet results are good, while the authors could provide more evidence to show its strength and how to balance the computation and model performance. For the experimental setup, it is good and reproducible. However, when digging deeper, the reason for the ensemble is that we want to find a way to calculate the features through different classifiers, maybe this is because a single classifier is not able to learn all the features from the images at once. But why it is necessary to use this approach (it also needs pre-training) instead of using a more powerful network to achieve a similar performance?

It is really good to see those analyses on Single Encoder multiple Representation, Multiple Encoder Multiple Representation, and Selection of Encoders and Representations for the Ensemble. It would be suggested if the author could give a detailed interpretation of the selected layer and how it could be used in other settings.

This paper is well-written, with not many typos.

The topic is inspiring and interesting while it is not clear how the ensemble could help FSL tasks. The improvement is not obvious and the results are not enough. Also, it would be better the authors could provide more analysis about why this ensemble works. It would be better the authors could give an analysis of the hyper-parameters of the proposed method. For example, in 5.3 Selection of Encoders and Representations for the Ensemble, τ = 0.93, but how the model performs when τ is different and how we could find an appropriate τ when doing ensemble. The authors should provide enough support to justify the validity of the methods and why this method is worth doing in comparison with other methods. Also, it is worth discussing other aspects such as flops, params, etc.

---

### Official Review · AnonReviewer3 · 2020-10-30
**Good results, but the idea is not novel**

**Rating:** 4
**Confidence:** 5

**Review:**

Summary:
The authors propose to tackle the problem of few-shot learning (FSL) using ensembling diverse classifiers. The diverse classifiers are obtained using the outputs from different intermediate layers of a pre-trained CNN feature extractor (or multiple CNNs). As a result, the authors demonstrate state-of-the-art accuracy on two mini-ImageNet and tiered-ImageNet datasets.

Pros:
- The idea totally makes sense since, in few-shot learning, the test distribution may be quite different from the training one. Hence, employing lower-layer features that are more class-invariant must be helpful, even though the space of semantic concepts learned by earlier layers is probably not as reach as for the deeper layers.
- The results in mini-ImageNet and tiered-ImageNet are impressive. The experimental section is informative and clear.
- The paper is well written and is easy to follow.

Cons:
- Limited contribution. None of the introduced ideas in this paper is novel. For example, the idea of using ensemble methods for FSL was introduced in [1]. Then, the idea of aggregating information from intermediate layers of a feature extractor to build a reacher classifier for FSL was introduced in [2]. The authors of [3] also used intermediate layers for better classification results. Basically, the contribution of the current work is to combine the ideas of [1] and [2] while using a different backbone network (a new ResNet18) and a different classifier (RelationNet).
- I would call the need to manually select the layers from which to build classifiers a down-side of the approach since selecting all representation would lead to degraded performance.

Overall, I like how the paper reads. However, the contribution of this work boils down to combining existing ideas and methods into a new pipeline, which I don't find sufficient for the ICLR acceptance standard.

[1] - Dvornik et.al. "Diversity with cooperation: Ensemble methods for few-shot classification"
[2] - Dvornik et.al. "Selecting Relevant Features from a Multi-domain Representation for Few-shot Classification"
[3] - Rusu et.al. "Meta-Learning with Latent Embedding Optimization"

---

### Official Review · AnonReviewer2 · 2020-10-30

**Rating:** 4
**Confidence:** 4

**Review:**

This paper presents a deeply supervised few-shot learning model via ensemble achieving state-of-the-art performance on mini-ImageNet and tiredImageNet. The authors first studied the classification accuracy on mini-Image across convolutional layers and found the network could perform well even in the middle layer. Therefore, they added classification headers on the selected layers, so that these layers can directly output predictions. The final result is the ensemble of all the select layer predictions, called the Multiple Representation Emsemble. To improve the result, they further average the results of two models with different network backbones, called Multi-Model Emsemble. The results show this method can achieve state-of-the-art performance on the two datasets.

Advantage:
1. The motivation and idea in this paper are clear and simple, so the reader is easy to understand it.
2. Figures 2 and 3 are nice, which are clearly demonstrate the motivation and algorithm.
3. The find in Figure 2(a) is very interesting. The middle layer has a better representation than the end on the few-shot image classification task.
4. The results are positive.

Disadvantage:
1. The idea in the paper is not very novel. The main contribution of this paper is doing a deep supervision ensemble. However, people have studied deep supervision learning for a while on image classification [1], segmentation [2], and depth estimation [3]. Specifically, [2] [3] also fuse the multi-layers' outputs.
2. The authors only show the ensemble results via averaging scores over the models. It will be good to study more ensemble methods. For example, the deep layer has higher accuracy than the shallow layer. Is it possible to assign a different ensemble weight for each layer based on the accuracy?
3. In figure 2(a), why the middle layer performs better than the last layer? It will be good to show some analysis?
4. In table 1, since the proposed model has done a model ensemble, it cannot directly compare with CAN and CTM. Should add the result without ensemble in table 1. If I put the third-row result "64.03" in table 2 to table 1, the improvement would be marginal.
5. Both mini-ImageNet and tired-ImageNet are the subsets of ImageNet. To verify the generalization, it will be good to add CIFAR, meta-iNat [4], or CUB [5] results.

Minor mistakes,
1. Equation 1, should add the superscript `n` to r.
2. Figure 1, the characters are not evenly spaced.
3. Figure 2 (a), the axis label is too small.
4. In section 4.1, the sentence "The model can be pre-trained ......Dtrain or Dval.)" is redundant, which is common sense.
5. In section 5.1, "After pre-training, we added shift and scaling parameters for the convolutional layers in the encoder and trained the parameters by the MTL approach used in". Might add more details about the shift and scale, so that the reader does not have to read another paper.
6. Table 1, the standard deviations in "our" results are not aligned.

----- post rebuttal ----
The authors haven't addressed my questions. I would keep my score unchanged.
One more comment: I suggest the authors compare to a related baseline SimpleShot [6] that is arguably less complicated.

Overall, given that the novelty and improvement are minor, I think this paper might be not ready at this time.

[1] Lee, Chen-Yu, et al. "Deeply-supervised nets." Artificial intelligence and statistics. 2015.

[2] Xie, Saining, and Zhuowen Tu. "Holistically-nested edge detection." Proceedings of the IEEE international conference on computer vision. 2015.

[3] Chang, Jia-Ren, and Yong-Sheng Chen. "Pyramid stereo matching network." Proceedings of the IEEE Conference on Computer Vision and Pattern Recognition. 2018.

[4] Wertheimer, Davis, and Bharath Hariharan. "Few-shot learning with localization in realistic settings." Proceedings of the IEEE Conference on Computer Vision and Pattern Recognition. 2019.

[5] Wah, Catherine, et al. "The caltech-ucsd birds-200-2011 dataset." (2011).

[6] Wang, Yan, et al. "Simpleshot: Revisiting nearest-neighbor classification for few-shot learning." arXiv preprint arXiv:1911.04623 (2019).

---

### Decision · Program_Chairs · 2021-01-07
**Final Decision**

**Decision:**

Reject

**Comment:**

This paper introduces an ensemble method to few-shot learning.
Although the introduced method yields competitive results, it is fair to say it is more complicated than much simpler algorithms and does not necessarily perform better. Given that ensembling for few-shot learning has been around for a while, it is not clear that this paper will have a significant audience at ICLR.
Sorry about the bad news,

AC.